# Vaccine Vectors Harnessing the Power of Cytomegaloviruses

**DOI:** 10.3390/vaccines7040152

**Published:** 2019-10-17

**Authors:** Mario Alberto Ynga-Durand, Iryna Dekhtiarenko, Luka Cicin-Sain

**Affiliations:** 1Department of Vaccinology and Applied Microbiology, Helmholtz Centre for Infection Research, 38124 Braunschweig, Germanyiradekh@gmail.com (I.D.); 2German Centre for Infection Research (DZIF), partner site Hannover/Braunschweig, 38124 Braunschweig, Germany; 3Centre for Individualised Infection Medicine (CiiM), a joint venture of Helmholtz Centre for Infection Research and Hannover Medical School, 30625 Hannover, Germany

**Keywords:** cytomegalovirus, vaccine vector, memory inflation, immunotherapy, immunization, vaccine development

## Abstract

Cytomegalovirus (CMV) species have been gaining attention as experimental vaccine vectors inducing cellular immune responses of unparalleled strength and protection. This review outline the strengths and the restrictions of CMV-based vectors, in light of the known aspects of CMV infection, pathogenicity and immunity. We discuss aspects to be considered when optimizing CMV based vaccines, including the innate immune response, the adaptive humoral immunity and the T-cell responses. We also discuss the antigenic epitopes presented by unconventional major histocompatibility complex (MHC) molecules in some CMV delivery systems and considerations about routes for delivery for the induction of systemic or mucosal immune responses. With the first clinical trials initiating, CMV-based vaccine vectors are entering a mature phase of development. This impetus needs to be maintained by scientific advances that feed the progress of this technological platform.

## 1. Initial Considerations on Cytomegalovirus as a Viral Vector

Cytomegalovirus (CMV) belongs to the β-herpesvirus subfamily that includes roseoloviruses, such as human herpesvirus 6 or 7 and numerous CMV variants that coevolved with diverse mammalian hosts, including human CMV (HCMV), murine CMV (MCMV) and rhesus CMV (RhCMV). CMVs are characterized by very large DNA genomes, usually asymptomatic infection in healthy hosts [1] and the strongest cellular immune response known in clinical medicine [2]. As a human opportunistic pathogen that causes severe disease in immunodeficient individuals, HCMV biology has been extensively studied. CMV infection is strictly species-specific and restricted to its natural host [3]. Nevertheless, experiments in animal models infected with the orthologous virus have proven useful in advancing the understanding of HCMV infection, pathogenicity and immunity [4]. Two shared cardinal features of CMV infection are (i) viral persistence in absence of overt replication but associated with reactivation capability defined as “latency” [5,6] and (ii) the induction of a large pool of functional antigen-specific T cells in the periphery that follow expanding kinetics [7,8,9]. This latter phenomenon is referred as “memory inflation” (MI) due to the fact that CMV-specific T cells accumulate over time [10], comprising on average 10% of CD4 and CD8 T-cell memory compartments in humans [2]. While the slow growth of HCMV in tissue culture may present obstacles for the industrial scale production of CMV-based vaccines, the large pool of functional antigen-specific T cells in the periphery and the possibility to modify CMV genomes due to improvements in viral genetics [11] draw a strong interest to CMV as a vaccine vector [12,13] (Table 1). The inflationary response appears linked to immune protection [14,15] and has been proposed as a target for immunization induction [16].

Besides the exceptional induction of adaptive immune response in humans [2,37], CMV possesses assets that conveniently address weaknesses common to other vector candidates. The cloning of CMV genomes as bacterial artificial chromosomes in *Escherichia coli* [38] has allowed genetic engineering approaches to effectively express multiple exogenous immunogens or modify large genome portions [39,40]. Furthermore, CMV is known to superinfect hosts with a history of prior exposure to the virus [41] due to its immune evasive properties that protect the virus from recognition by primed T-cells [42] and CMV vectors induce protective immunity in experimentally vaccinated animals with documented prior exposure to CMV [18,19,20,36]. Therefore, CMV vector candidates would avoid immune interference as described for AdV. CMV is a pathogenic organism in immunodeficient populations [43] or in congenitally infected children. Therefore, it is imperative for any HCMV vaccine vector to be attenuated. The generation of replication deficient CMVs that maintain their immunogenicity [36,44] and the modification of genome portions that contain evasions genes [45], as well as the insertion of activating ligands to increase immune control of CMV [46,47,48] are strategies that will be described in this review. Interestingly, a replication-deficient MCMV carrying an attenuating mutation within the DNA primase gene, while able to induce primary T cell responses could not replicate the characteristic MI CD8 response [49]. This finding indicates that a limited viral replication or expression of viral transcripts may be required for induction and maintenance of the particular CMV immune responses.

Research with “single cycle” vectors has been particularly helpful to extend these observations. In MCMV−ΔM94, the essential gene *M94* was deleted, resulting in a non-pathogenic MCMV that was unable to spread from the initial infected cell. Nevertheless, MCMV-ΔM94 still induced protection against an exogenous antigen encoded in the vector, ovalbumin (OVA), with a humoral and cellular response similar to WT MCMV [50]. Another single cycle mutant MCMV was generated by deleting the essential glycoprotein L (ΔgL−MCMV), resulting in MI induction after systemic administration and similar immunodominance hierarchy [44,51]. Similarly, a single-cycle RhCMV with the *gL* gene deletion was engineered [52] that elicited RhCMV-specific CD8 T cell responses upon immunization but no protection against RhCMV challenge. It is unclear if differences in immunization outcomes between single-cycle RhCMV and MCMV reflected underlying differences between primate and rodent CMVs but they do argue that separating virus replication and pathogenicity from its immunogenic capacity remains a challenge that requires better understanding of CMV in vivo replication, persistence and antigen expression. Non-propagating CMV vectors’ development is pivotal to prevent risks associated to vector reactivation and transmission. Evidence of reactivation after immunosuppression and posterior reversion to virulence of live attenuated CMV vaccines has been described since early exploratory studies [53] and constitutes a critical issue to be addressed. Recently, a RhCMV vector lacking the *Rh110* gene (RhCMVΔRh110), which encodes the tegument protein pp150 was shown to combine strong in vitro growth with low in vivo replication and pathogenicity [23], yet elicited strong and protective cellular immune responses against simian immunodeficiency virus (SIV) challenge [17,24]. Remarkably, urine, saliva or breast milk shedding was not sufficient to mediate transmission between immunized mothers and naïve infants despite close contact. Even more significantly, this attenuated RhCMV vector was not able to reactivate and disseminate despite viral transfer in myeloid cells to naïve subjects. A recombinant MCMV expressing a high affinity immune receptor ligand (MULT-1MCMV) is another example of drastic attenuation with immunogenicity preservation [47]. Despite viral control, MULT-1MCMV was able to induce a protective CD8 T cell response against a vectored epitope. Notably, this strong attenuation was maintained even in immunologically immature newborn mice. These examples suggest that is possible to uncouple the ability of attenuated CMV to cause disease while maintaining their vector advantages and will be further discussed on this review.

The role of CMV as an oncogenic virus has not been completely clarified. HCMV infection is associated with some conditions that promote cell transformation, because several HCMV proteins disrupt the cell cycle upon infection [54] or activate the telomerase gene expression [55]. HCMV mediated mutagenesis [56,57], inactivation of tumour suppressors [58], immunoevasion of infected cells [59] and subsequent development of an immunosuppressive microenvironment [60] partially overlap with phenomena promoting the immune evasion by cancer cells [61]. Some clinical studies have suggested a role for CMV in the pathogenesis of glioblastoma, an aggressive neoplasm of the central nervous system [62,63] but subsequent studies have shown conflicting results concerning HCMV identification in tumours [64,65,66,67,68], a fundamental requisite according to the criteria proposed by Frederick and Relman for the identification of diseases caused by viruses [69,70]. Therefore, a consensus to consider CMV as an oncogenic virus or merely an oncomodulatory virus that may cause tumour cells to become more aggressive has not been reached [71]. On the other hand, CMV infection may have anti-tumour effects related to an increased antitumoral immune response [72]. CMV may induce a proinflammatory state that leads to antitumoral cellular responses [73]. It has been proposed that these conflicting findings may be reconciled by a model in which the state of CMV is critical for the final effect on tumour growth [74]. According to this model, latent CMV does not induce antitumoral responses, while an active infection may influence tumour environment and repress tumour growth. As CMV vaccine vectors are being developed to target cancer, more experimental evidence will be needed to characterize CMV effect on neoplasia by itself and how this situation may affect further vaccine design.

Finally, the exquisitely strong immune response to CMV infection raises biosafety concerns if one was to use it as a vector. The infection with HCMV is a major non-heritable factor shaping the inter-individual variation of the human immune response [75]. Whether this relationship is advantageous or detrimental to the immunocompetent host is debatable. On one hand, latent MCMV infection may compromise immune function in aging hosts according to some [76] but not all studies [77]. On the other, MCMV latency could have a beneficial effect on the immune response against bacterial pathogens [78]. Surprisingly, HCMV–infected healthy young adults developed stronger immune responses to the influenza vaccine, suggesting favourable effects of CMV in immunocompetent individuals [79]. In short, while the jury is still out on the immunomodulatory effects of CMV infection, CMV remains a highly interesting vector platform due to its genome engineering flexibility, a superinfection capacity that overcomes pre-existing immunity and the development of attenuated vectors that improve their safety profile.

## 2. Innate Immune Response to CMV is Relevant for Its Vector Properties

Viral vectors, replicative or not, are recognized by multiple host innate immune pathways. For this reason, viral vectors have been broadly described as “self-adjuvanted” [80]. Early inflammation prompts the subsequent development of adaptive immunity and is a powerful driver of vaccine response as demonstrated by the use of adjuvants in successful immunization strategies [81]. Notably, viral vectors do not require such help even after attenuation, as the initial response is sufficient to reach the threshold of immunogenicity. The paradigmatic replication-deficient vector Modified Vaccinia Ankara (MVA) strain has shown an immunogenicity profile that is equal or greater than VV replication-competent strains [82], because MVA drives stronger type I IFN responses [83]. Although it is close to impossible to study early stages of HMCV infection, MCMV infection models give valuable insights for understanding multiple features of immunity of its human counterpart. This model distinguishes three different response phases [84]—the innate and adaptive stages, both typical stages for most infections and immunizations, plus a MI response associated to viral latency. It is critical to understand how the early interaction between a CMV vector and its host may influence the induction of protective immune response, as it will be required for the vector to escape initial control and at the same time avoid pathogenicity. In the first phase of MCMV infection, type I IFN and Natural Killer (NK) cells have the most important role in containing propagation [85,86]. Initial IFN production takes place within the first 8 h in a cGAS/STING dependent manner in liver Kupffer cells [87] and in splenic stromal cells [88]. This early IFN wave restricts viral production through NK cells activation and orchestrates posterior CMV control [89]. CMV encode genes that manipulate the host immune response by engaging inhibitory receptors and suppressing activating signals on NK cells [90] or interfering with co-stimulating receptors expression on myeloid cells [91,92]. This set of products have been termed immunoevasins [90]. We have explored the capacity to provide functional and protective immunity with a replication-competent MCMV recombinant lacking genes that modulate the immune response. Thus, we deleted viral genomic regions rich in immunoevasins that target major histocompatibility complex class I (MHC-I) or their co-receptors, as well as NK-cell ligands. The resulting mutant induced cellular and humoral immunity and protected mice from WT MCMV infection [45], arguing that interference with viral immune evasion is a viable strategy in developing viral vaccine vectors. In this regard, MCMV gene products known to inhibit the expression of ligands of the activating NK cell receptor NKG2D have attracted considerable attention. These MCMV immunoevasins include *m145* (inhibits mouse UL16-binding protein-like transcript (MULT)-1) and *m152* (inhibits RAE-1γ surface expression) [93]. Importantly, NKG2D is a potent activating receptor expressed also on NKT cells, γδ T cells and CD8 T cells. CMV vectors with immunoevasins deletion simultaneously expressing NKG2D activating ligands enhance NK cells viral control and display attenuated in vivo fitness and virulence [94]. As a model of this stratagem, a recombinant MCMV expressing RAE-1γ (RAE-1γMCMV) in place of the *m152* gene was dramatically attenuated in vivo in both immunocompetent and immunodeficient hosts and elicited a protective and long-lasting CD8 T cells [46]. These results support the concept that NK-mediated attenuation does not compromise specific immune response, while RAE-1γ mediated NKG2D engagement on CD8 T cells may provide co-stimulatory signals that enhance their responses [95]. In fact, a RAE-1γMCMV expressing an immunodominant CD8 T-cell epitope of *Listeria monocytogenes* listeriolysin O or OVA-derived SIINFEKL epitope induced specific and protective CD8 T cell response to both antigens. Unexpectedly, this response was also preserved in NKG2D^−/−^ mice, suggesting an NKG2D-independent immune function of RAE-1γ [28]. Consequently, a RAE-1γMCMV vector expressing the SIINFEKL epitope induced a robust CD8 T cell response able to contain the growth of an OVA-expressing tumour in prophylactic and therapeutic settings [48]. Therapeutic RAE-1γMCMV vaccination was further enhanced by blocking the immune checkpoints TIGIT and PD-1, supporting a synergistic role with tumour immunotherapy. Functionally, RAE-1γMCMV induced a dominant population of KLRG1 + SIINFEKL-specific CD8 T cells that were able to secrete effector cytokines. KLRG1 marker with conserved functionality correlates with the MI phenotype [96]. Recently, a recombinant MCMV expressing another NKG2D ligand inserted in the place of its viral inhibitor *m145* was produced (MULT-1MCMV) [47]. MULT-1 has the highest affinity for the NKG2D receptor among all mouse NKG2D ligands. Attenuation of this mutant was even stronger than RAE-1γ-MCMV, yet MULT-1MCMV induced an efficient CD8 T cell response to viral epitopes and the vectored SIINFEKL epitope. MULT-1MCMV and RAE-1γMCMV results demonstrate that manipulation of MCMV susceptibility to the innate immunity may be efficiently used to generate attenuated but still immunogenic vaccine vectors. Remarkably, an HCMV expressing ULBP2, an NKG2D ligand, was susceptible to NK cell control and preserved its ability to stimulate specific T cells [97], demonstrating that this strategy may be used in HCMV vector candidates.

It is worth mentioning a remarkable example of nonspecific innate protection using a recombinant MCMV vector expressing the *Mycobacterium tuberculosis* (Mtb) mycolyl transferase 85A antigen (MCMV85A). Protection after Mtb challenge was found to be comparable to the benchmark vaccine, bacillus Calmette–Guérin (BCG) but surprisingly, low levels of specific CD8 and CD4 responses after immunization were found [29]. Mycobacterial load reduction was shown to be dependent on NK cell responses through IL-21 signalling with a minor role of specific T cell immunity. A similar innate mechanism has been associated to BCG-induced protective immunity against Mtb [98]. Further studies exploring the protective effect of CMV vectors by its induction of innate immunity are required.

## 3. Adaptive Immune Response to CMV Relevant for Its Vector Properties

Long-lasting CMV control cannot be achieved by innate immunity alone, because it requires the establishment of an effective adaptive immunity. In immunodeficient individuals with acquired losses of their adaptive immunity, CMV causes a disease with high morbidity and mortality [1].

### 3.1. Humoral Response

HCMV induces potent specific antibodies but their role in controlling CMV infection is controversial. Reports on polyclonal IgG efficacy in congenital CMV infection [99,100] and in CMV prophylaxis in transplantation [101,102] showed very divergent outcomes. This apparent contradiction might be interpreted in light of a recent evidence showing that only strain-specific antibodies are effective in preventing MCMV reactivation in a model of post-transplant infection [103]. Evasion of CD8 T cells by virally-encoded MHC-I inhibitors further supports CMV replication and dissemination at superinfection [42]. Taken together, these data offer a mechanistic explanation of how superinfection with multiple genetic variants of HCMV is possible [104]. These considerations are highly relevant for CMV vectors, where pre-existing immunity to one CMV strain does not inevitably confer protection against other strains [105] allowing the use of this vector in CMV-seropositive populations. IgG antibody inflation in MCMV infection was observed and high-dose infection was associated with the presence of neutralizing antibodies [106]. Importantly, an “inflationary” B cell response that gradually increases IgG levels to high frequencies was found in the persistent phase of HCMV infection [107]. This may indicate that CMV vectors may be exploited to induce not only an expanding T-cell response but also a life-long humoral immune protection against vectored antigens. Nonetheless, antibody responses induced by CMV vectors against heterologous expressed target antigens remain relatively uncharacterized. In an MCMV-based immunocontraception setting to prevent mouse plagues, the expression of the zona pellucida 3 (ZP3), a self-antigen from the egg and a primary sperm receptor, induced sterility in mice [108]. Autoimmune ovarian damage and infertility was demonstrated to be antibody-mediated [109], establishing the proof that humoral effector responses may be induced. Similarly, a single dose of an MCMV-based tetanus vaccine (MCMV/TetC) expressing the non-toxic fragment C of the tetanus toxoid (TT), induced protective levels of anti-TT antibodies. The antibody mediated protection induced by MCMV/TetC was more durable than the alum-adsorbed detoxified tetanus vaccine, considered to be the gold standard [110]. Although originally constructed for T cell immunity induction, an RhCMV-based Ebola virus vaccine expressing full length GP (RhCMV/EBOV-GP) resulted in immune protection against Ebola virus challenge, with substantial levels of GP specific antibodies and no detectable CD8 T cell responses [25]. Interestingly, RhCMV/EBOV-GP-induced antibodies did not have neutralization capacity, implying a non-direct antiviral mechanism of action such as antibody-dependent cellular cytotoxicity or complement activation. In a B16 melanoma model, MCMV-TRP2, a recombinant MCMV carrying an unmodified melanoma antigen (mouse tyrosinase-related protein 2 TRP2), induced prophylactic protection mediated by antibodies [111]. This protection was also conferred by a spread deficient version lacking the essential glycoprotein gL (ΔgL-MCMV-TRP2), which is in stark contrast to a spread deficient RhCMV that failed to induce neutralizing humoral responses or provide protection [52]. Therefore, a better understanding of humoral response induction by non-propagating CMV vectors is required to evaluate their potential for practical applications.

Humoral response is relevant for CMV vectors not only as an effector component against a targeted antigen but also as a potential pathogenic mechanism. Unexpected possible side effects are associated with higher reactogenicity due to antibody-mediated activation [112] or even increased susceptibility to the targeted disease, as it was the case with AdV vectors against HIV [113]. Concerns related to autoimmunity induction by HCMV infection, as proposed by the discovery of an antibody against the HCMV pp150 protein that cross-reacts and eliminates CD56^bright^ NK cells [114] underline the necessity of further studies. In sum, CMV vectors have the potential to induce an efficient humoral response in certain setups but additional research and understanding of this immune arm is necessary to address safety considerations.

### 3.2. Cellular Response—T cells

As previously described, MCMV replication in most organs is initially controlled in the course of the first week while specific T cell are primed. After antigen clearance, virus-specific memory T cells are preserved and, in contrast to naïve T cells, rapidly respond upon re-encounter with a cognate antigen through direct effector functions or further proliferation and differentiation into a more advanced phenotype [115]. Memory subsets have been categorized, according to their functions, location, migration abilities and maintenance requirements into central memory T cells (TCM), effector memory T cells (TEM), tissue-resident memory T cells (TRM) and peripheral memory cells (TPM) [116]. CMV infection, with its lifelong intermittent antigen expression in latency, results in prolonged antigen exposure that impacts T cell responses [5]. Besides the well-described typical pattern of expansion and contraction of antigen-specific T cells, CMV infection is characterized by memory inflation [8], a virus-specific T cell response with skewed specificity that continues to expand during viral persistence [9,117]. This “inflationary” population is represented by circulating TEM (CD62L^−^) which in contrast to TCM (CD62L^+^) have a phenotype compatible with recent antigen activation that is believed to be induced by low level viral persistence through sporadic reactivation events [96,115]. Unlike other persistent infections such as HIV, hepatitis C virus (HCV) and hepatitis B virus (HBV), CMV infection leads to the production of fully functional, virus-specific T cells without any hallmarks of immune exhaustion [118,119]. Thus, current MI definition integrates these findings in three cardinal characteristics [7] (1) Restricted contraction after priming, with a durable memory pool, (2) Dominant and sustained effector phenotype and (3) No signs of exhaustion. Adoptive transfer experiments in mouse models showed that MCMV-specific TEM have a half-life of 45 days to 12 week [96,120] and are differentiated from TCM during viral reactivation with no need of new thymic emigrants [121]. The localization where the presumed reactivation events and TCM activation takes place is still a matter of debate. Two non-conflicting models [122] are proposed—one where the site of activation is located in the lymph nodes [123], while other emphasize on circulating cells stimulation by endothelial cells [124]. In either of the models, antigen presentation is performed by non-hematopoietic cells [123,125] and thus not dependent on the immunoproteasome [126]. These observations in MCMV mirror those found in HCMV infection, as HCMV induces the expansion of selected epitope-specific and stable TEM [10]. This expansion, while much more variable than in MCMV, has been designated as HCMV-induced MI, because both share the same broad features of no-contraction and functionality. As an example of common characteristics, MCMV and HCMV-induced MI populations use distinctive co-signalling pathways for their activation as they lack CD27 and CD28 expression [121,127]. Most evidence regarding MI induction and protection comes from CD8 T cell analysis. While high numbers of HCMV-specific CD4 T cells displaying a cytolytic phenotype are observed after infection [128] and MCMV CD4 T cell epitope vaccination have been shown to protect against WT infection [129]. CD4 inflation has been much less studied. In a seminal work, a CD4 T cell response specific for the MCMV m09^133–147^ peptide was found to be inflationary (although the size of the response was less impressive than those observed in CD8 inflation), indicating that CD4 T cell responses are diverse and may be an interesting target for immunization [117]. This will be expanded on the section dedicated to CD4 T response.

Data from the single-cycle MCMV model argue that the majority of cells that differentiate into an MI phenotype are primed during the acute phase of infection [44]. The magnitude of response during this priming defines the size of MI [130] but latency is required to maintain MI population [44,96]. For the purpose of understanding MI responses, we should look into the nature of CMV latency [5,131]. Latency refers to a stable, nuclear and extrachromosomal maintenance of the viral genome for the life of the host with a restricted gene expression and a lack of virus production. CMV can establish latency in several cell types [132]. MCMV has been shown to establish latency in stromal cells, as those that line spleen sinusoids [133] and cells co-localizing to endothelia in a variety of organs [134]. In human infection, CD34+ progenitors and their derivatives, including granulocyte–macrophage progenitors and CD14+ monocytes, have been associated with HCMV latency [135]. The definition of latency includes the capacity of reversion from the dormant state to a full replicative potential capacity, a phenomenon named reactivation. Intermittent abortive reactivation events allow low levels of viral transcription that are believed to be the main driver of MI, as the “immune sensing hypothesis” states [136]. MI likely equilibrates viral genomic activity, restricting CMV cycle progression, as epitopes derived from immediate-early (*ie*) genes act as checkpoints that control downstream transcriptional activity [137]. Recent single-cell transcriptomic data supports the model of stochastic transcription of HCMV antigens in latent non-hematopoietic cells [138]. This data is consistent with a model of vibrant equilibrium between host and virus, where inflationary T cells have a protective and balancing function that keeps the number of viral reactivation events in check. Moreover, directed depletion of inflationary specific population is followed by a subsequent rapid recovery, expansion and conserved functionality of inflationary T cells, evidencing MI as the host’s “immune sensing” counterpart to CMV activity [139]. This is relevant for vector design, as continuous Ag-specific immune cell proliferation is one of the defining factor for maintenance of protective responses to vectored antigen, next to conserved functionality.

At this point, is important to remember that not all CMV immunodominant epitopes induce inflationary responses. Since MI induction is the one of the key reasons to consider CMV as a vaccine vector, it is essential to identify the factors that define if an epitope will induce inflationary responses. Gene expression, epitope avidity and peptide processing are main determinants of MI induction [16]. Out of these determinants, the most intuitive one is gene expression. Operationally, CMV viral transcripts have been divided into three classes [140]. Immediate-early (*ie*) RNA (identified in the first 6 h post infection), early RNA (synthesized up to about 24 h after infection and required viral transactivators [141]) and late RNA (synthesized after 24 h, requiring viral DNA replication). Low levels of *ie* genes are expressed in latently infected cells and well-recognized inflationary epitopes are encoded by these genes [142,143]. Similarly, the HCMV *ie1* gene expresses immunodominant HLA-I restricted epitopes [144]. Hence it is not surprising that a recombinant MCMV expressing an epitope from nucleoprotein (NP) of influenza A (IAV) virus in the context of the *ie2* gene (MCMV-NP) induced inflationary responses [39]. Influenza virus infection induced a 20-fold higher primary CD8 T-cell response than MCMV-NP but this response contracted later and was not associated with protection. MCMV-NP protective immunity was weak initially but improved over time. A similar strategy was used to generate MCMV vectors expressing an epitope derived from the NP gene of Zaire Ebolavirus (ZEBOV) in the *ie2* gene (MCMV/ZEBOV-NPCTL). This vector also induced high levels of long-lasting CD8 T cells against the *ie2* encoded epitope and robust immune protection against ZEBOV challenge [26,27] comparable to the well-established VSV-based vaccine VSVΔG/ZEBOVGP [145]. Neutralizing antibody activity was not detected in any convalescent serum from MCMV/ZEBOV-NPCTL vaccinated mice demonstrating that a mechanistic role of humoral immunity was minimal in this system.

While these studies confirmed that *ie* expression in MCMV is associated with MI and effective immune protection, it is not excluded that genes expressed later in the cycle may also be targeted for antigen expression and MI induction. Therefore, our group generated MCMV mutants carrying the same epitope in genes with different expression patterns. Using MCMV recombinants expressing a single CD8 T cell immunodominant epitope from HSV-1 fused to different MCMV genes, we demonstrated that the pattern and magnitude of T cell responses against an epitope depends on its gene-expression context. As previously described for other epitopes [39], HSV-1 epitope fusion to the *ie2* gene resulted in inflationary response to the exogenous epitope [33]. In contrast, insertion of the HSV-1 epitope in the context of the *M45* gene, an early gene, was associated with an early expansion followed by contraction. Interestingly, both vectors induced TEM responses [14] and protective immunity, which was confirmed in an HSV-1 challenge experiment [33] or upon challenge with a recombinant VV expressing the HSV-1 epitope [15]. Another set of recombinant MCMVs expressing the low-avidity H-Y derived epitope [146] in either *ie2* or *M45* gene locations showed a similar pattern of inflationary responses that were restricted to the *ie2* gene expression context [15], although at lower levels (Figure 1). However, in this case, only the *ie2* expression context resulted in protective TEM against a recombinant VV expressing the same epitope [15]. Therefore, the TEM phenotype of responding cells was associated with immune protection in these studies. Another substantial implication for vaccine design coming from this study was that even low-avidity epitopes may provide immune protection if expressed in an inflationary context.

In the case of MCMV vaccines against tumours, even optimal antigen expression may not be sufficient to provide protection against metastatic cancer. On the other hand, MCMV vectors carrying modified epitopes with increased antigenicity were shown to induce immune responses that are sufficient to provide protection in therapeutic settings. For instance, the MCMV expressing the B16 melanoma antigen gp100 (MCMV-gp100KGP) induced a strong, long-lasting gp100-specific CD8 T-cell response with a durable, polyfunctional phenotype with inflationary kinetics [147]. The modified gp100 (gp100KGP) expressed by MCMV induces stronger cytotoxic T lymphocyte responses [148] than an MCMV vector with the original gp100 epitope. Prophylactic and therapeutic MCMV-gp100KGP immunizations protected mice from tumour induction and progression respectively through gp100-specific CD8 T cells. MCMV-gp100KGP protected the mice better than a VSV vector expressing the modified gp100 peptide (VSV-gp100KGP). Therefore, therapeutic vaccines relying on a combination of selected antigens and adequate expression context may provide sufficient immune protection to tackle neoplastic targets.

CMV vaccine vectors encoding whole antigenic proteins were explored with variable success. Advantages of using a full-length protein include the elicitation of broad immune responses in various MHC backgrounds which increases the prospect of a protective response toward critical epitopes in genetically heterogeneous populations [149]. A comparison of MCMV-based vectors encoding either the full length gene of the prostate-specific antigen (PSA) (MCMV/PSA_FL_) or a PSA-derived epitope (MCMV/PSA_65–73_) within the *ie2* gene [150] showed that only MCMV/PSA_65–73_ protected against tumour challenge with a PSA-expressing adenocarcinoma, although both vectors induced an inflationary response of PSA-specific CD8 T cells. As MCMV/PSA_FL_ induced poor CD8 response upon challenge, the mechanism of CD8 suppression was suspected to depend on differences in the functionality of responding cells. One potential explanation for such discrepant outcomes relates to epitope availability to proteasomal processing, which will be described in the following section.

### 3.3. Antigen Processing in MI

Differences in protection between whole-protein versus single-epitope recombinant CMV in PSA model were replicated in a human papilloma virus strain 16 (HPV16) model [14]. Here, an MCMV expressing only a class I epitope induced better protection against tumour challenge than the vector encoding the full-length E6 and E7 proteins. This discrepancy coincided with the vector capacity to induce inflationary responses, which were robust against the vector with the single epitope and poor when the epitope was expressed from the full-length protein, although the two vectors expressed the protein (or the peptide) from the very same viral promoter [14]. This difference is explained by peptide availability to proteasomal processing. The proteasomes are the major protein degradation machinery and are also responsible for the processing of antigens for presentation by the MHC class I pathway [151]. Therefore, altering amino acid residues flanking an epitope boosts peptide processing and T cell responses in CMV vectors [152]. A specialized type of proteasomes, named the immunoproteasome, is constitutively expressed in professional antigen presenting cells and inducible in other cells by exposure to IFN-γ. Peptidomes generated by the constitutive proteasome or by the immunoproteasome are substantially different [153]. In our model, the epitope that was available to the constitutive proteasome induced inflationary and protective responses. If the same epitope, expressed within the same gene, required the immunoproteasome for processing, then it could not induce inflationary responses [14]. Likewise, in mice lacking LMP7, a critical subunit of the immunoproteasome, conventional T cell responses to MCMV are abrogated but MI is conserved [126]. Finally, MI is improved if the same epitope is flanked by amino acid residues that are more amenable for proteasomal processing [14]. Hence, epitopes available to the constitutive proteasome in latently infected non-hematopoietic cells induce MI and provide much stronger immune protection than the epitopes that are restricted to the immunoproteasome (Figure 2).

Another relevant driver in MI development is the competition among peptides [33]. The expression of heterologous antigens in the context of an *ie* gene reduces MI to endogenous MCMV peptides [15,33,154]. On the other hand, the deletion of an *ie* epitope from the MCMV genome increases latent transcription of downstream viral genes [137]. Deletion of immunodominant epitopes in a recombinant MCMV revealed a conditional immunodominant response to otherwise subdominant epitopes [155]. Therefore, epitopes expressed earlier repress the expression of later genes and thus outcompete epitopes expressed later in the virus cycle but only if they are expressed within the same latently infected cell. In line with this prediction, mice infected with a recombinant MCMV expressing the exogenous immunodominant epitope and co-infected with WT MCMV retained the MI to endogenous epitopes [154]. Namely, the transgenic epitope cannot interfere with the transcriptional activity of WT MCMV genomes that are latently maintained in another cell. Finally, only epitopes recognized by T-cells with high-avidity of T cell receptor (TCR) binding to peptide-MHC complexes (pMHC) successfully outcompeted the endogenous responses [15]. A low avidity epitope expressed within the *ie2* MCMV gene did not affect endogenous inflationary responses. Taken together, the evidence shows that competition for antigen lays at the level of the latently infected cell and determines MI antigen composition. Therefore, we have proposed that MI is in essence an epitope-competitive process influenced by their context of gene expression, antigen processing and avidity of TCR binding to pMHC [16]. The current model proposes the following—latent CMV in non-hematopoietic cells occasionally expresses latency-associated transcripts and proteins that are processed by the constitutive proteasome into epitopes. The high-avidity epitopes that are expressed earlier outcompete later ones and induce inflationary responses. Therefore, antigen processing and context of expression rise up as the defining tasks to be addressed by CMV vector design, as MI would provide an ideal target for predictable and potent immune protection (Figure 3).

### 3.4. Unconventional MHC Restriction

RhCMV vectors against SIV (RhCMV/SIV) encoding SIV Gag, Rev/Nef/Tat and Env induced large and durable specific T cell responses which protected 55% of animals against a particularly virulent strain in a SIV challenge [20]. Further analysis identified that protection was not associated with humoral response but correlated with SIV-specific CD8 T cells against “non-canonical” epitopes not described previously in SIV infection or vaccination [19]. Astonishingly, this atypical immunodominance involved CD8 T cells recognizing peptides presented by MHC II [21] and Mamu-E [22], the rhesus monkey version of HLA-E. This unexpected outcome depended on the vector backbone. The RhCMV vector was constructed using the fibroblast-passaged 68.1 strain, which lost several genes including *Rh157.4–157.6*, homologues to HCMV genes *UL128–131*. The products of these genes form pentameric complexes on the virion surface of RhCMV and HCMV respectively, determining viral tropism for several non-fibroblastic cells. Restoration of these genes in the RhCMV 68.1 resulted in a vector that elicited CD8 T cells restricted by classical MHC-I molecules. Responses against atypical epitopes were critical for immune protection against SIV challenge [156] but were dispensable for protection against other pathogens, such as Mtb [18]. Based on the ground-breaking RhCMV results, an HCMV that lacked the *UL128–131* genes was used in humans. Disappointingly, it failed to induce a similar MHC restriction pattern and immunodominance as RhCMV in rhesus monkeys [157]. Since *Rh157.4–157.6* were not the only genes missing from the RhCMV vaccine vector, the HCMV and RhCMV results may be reconciled by arguing that the deletion of pentamer genes from a CMV vector may not be sufficient to ensure the induction of unconventional T cell responses. Alternatively, this dichotomy may depend on differences between human and monkey immune response. The MHC complex is much more polygenic and polymorphic in rhesus monkeys than in humans [158]. Nevertheless, one should not dismiss the findings from the RhCMV model as irrelevant for the human situation. HLA-II restricted CD8 T-cell responses to natural HIV infections are rare but present in humans [159] and HLA-E-restricted induction of adaptive responses to infections is an exciting area of rapid advances [160,161]. Some HIV-derived peptides are loaded on HLA-E molecules [162] and the variability of HLA-E binding peptides seems to be due to structural characteristics of its peptide pocket malleability [163]. While their functionality is still debatable, HLA-E and HLA II epitopes are intriguing targets for improving vaccine design.

### 3.5. CD4 T Cells

Compared to CD8 T cell responses to CMV-encoded epitopes, CD4 CMV-specific T cells have been less studied, although they are relevant as their numbers inversely correlate to disease in various clinical CMV settings [164,165,166,167]. Based on HCMV immunity, it is known that that CD4 T cells help to sustain specific CD8 T cell populations in bone marrow transplants [168] and in MCMV they are required to control replication in the salivary gland [169]. With regard to MI, CD4 T cells develop a less dramatic accumulation in comparison to CD8 T cells. This may be due to differences in antigen exposure requirements, as CD4 T cells require a longer interaction, are more dependent on costimulatory signals and have less proliferative potential than CD8 T cells [170]. Besides their established function as modulators of CD8 T cell responses, CD4 T cells bear a direct and independent effector role. For instance, prime-boost subunit immunization of susceptible mice with a single CD4 T cell peptide epitope from Salmonella provided significant protection mediated through polyfunctional CD4 T cells, similar to a whole heat-killed Salmonella bacteria vaccine [171]. Importantly, no significant antibody or CD8 T response was found in these mice. The presence of similar effector CD4 T cells with a cytolytic phenotype expressing granzyme B and perforin is associated to MCMV infection [128] and to HCMV primary infection [129], therefore, is of great interest to vaccinologists. Cytotoxic CD4 T lymphocytes have been shown to play a protective role in several infections [172] including *M. tuberculosis* [173]. Thought-provoking data regarding the role of CMV vectors against Mtb comes from the rhesus macaque model. Four different RhCMV vaccines that together expressed nine different Mtb proteins were engineered [18]. Subcutaneous RhCMV vaccines (RhCMV/TB) administration induced TEM CD4 and CD8 T cell responses to all antigens. In comparison, BCG induced TEM CD8 and but no TEM CD4 T responses. CD4 induction in RhCMV/TB immunization resulted in a polyfunctional population able to produce both TNF and IFNγ. After Mtb challenge, RhCMV/TB was associated with diminished lung lesions and less extrapulmonary disease in comparison to BCG. These RhCMV vectors were engineered based on a 68-1 RhCMV backbone that was previously shown to induce unconventional (MHC-II and MHC-E restricted) CD8 T cell responses [21]. Remarkably, the main difference between the protective RhCMV/TB and the non-protective BCG immunization was the presence of polyfunctional, effector-differentiated CD4 T cell population. As a correlate of protection, neutrophil gene expression pattern predicted the protective outcome in RhCMV/TB-vaccinees, proposing an innate immunity effector arm associated to the vaccine, although a clear relationship between CD4 T cells and neutrophils has not been established. Apart from these relevant observations with regard to protection, two additional points related to safety issues should be noted. Since during MCMV infection the salivary gland contains IL-10 secreted by CD4  T cells [174], it has been suggested that the generation of a suppressive niche can be related to viral persistence. In this regard, CD4 T cells production of IL-10 has been shown to facilitate virus persistence in mucosal tissue increasing efficacy of horizontal transmission via mucosal secretions [175]. Based on this result from MCMV models, infective augmentation after HCMV immunization could be a possibility. Some of these concerns were alleviated by a cohort study in human volunteers, where no suppressive phenotype was found after lifelong HCMV carriage [176]. Another biosafety issue focuses on the relevance of viral coinfection and its impact on MI, specifically as HIV + CMV + individuals show inflation of CMV epitope-specific CD4 T cells [177]. The authors hypothesize that increased low-level antigen exposure from inflationary epitopes drives MI accumulative expansion and this could have a detrimental potential that has safety implication for CMV vaccines. Indeed, a recently identified CX3CR1 + CMV-specific CD4 T cells that target vascular endothelium may mediate CMV-mediated vascular damage [178]. In conclusion, we need to improve our understanding of the requirements for inducing inflationary CD4-T cell responses. On one side, CD4 T cells are exciting effectors with a broad range of functionality that includes direct cytotoxicity and indirect manipulation of the immune environment. On the other side, CD4 T cells are often associated with autoimmunity and immunopathogenicity. Therefore, even if a CMV vector does not specifically target CD4 T antigens, identification and silencing of endogenous potentially cross-reactive CD4 T cell CMV epitopes may improve the safety profile of CMV vaccine vectors.

## 4. Factor Affecting Immunity—Dose and Administration Route

Smart vaccine vector design may optimize antigen properties, peptide processing as well as timing of a transcript expression and thus ensure the induction of robust inflationary antigen-specific CD8 (and potentially CD4) responses. Although being a crucial step, vector optimization is not the final step towards optimal immune response- dose and administration route of vaccine vectors also critically influence their immunogenicity. Low dose viral inoculum, in contrast to high dose, resulted in reduced MI [179]. Since the induction and maintenance of high levels of effector-memory T cells is important for protective immunity in CMV vectors [36], this finding suggests that high dose immunization may be more convenient for vaccine purposes. Dose effect may be explained by a higher load of latent genomes that allows more transcripts to be expressed in latently infected antigen presenting cells. It is important to note that low dose viral inoculum elicits less MI but is associated with an improved secondary expansion capacity from the TCM population. This observation explains the ability of spread-deficient MCMV (ΔgL-MCMV) to induce MI, although at lower levels [44]. ΔgL-MCMV administration resulted in a relatively small number of latently or persistently infected cells but was still able to repeatedly activate the immune system and induce MI. Interestingly, pre-treatment with famcyclovir, a CMV antiviral that prevents initial amplification of the viral genome, abrogated MI [44], demonstrating the necessity of initial transcription activity after infection to allow MI.

Another critical factor that influences immunization efficacy is the administration route. Most models of CMV infection rely on systemic infection by intraperitoneal (IP) or intravenous route or by virus injection into the footpad in mice. A notable exception is the subcutaneous route, preferred for non-human primates [19,180]. Recent work has shed some light on the relationship between the administration route, CD8 immunogenicity and their protective effect. We showed that intragastric administration induces no MI, whereas intranasal infection induces less MI than IP infection [180]. Thus, systemic infection is the most effective route for inducing MI. Subsequent comparison of different routes of administration of MCMV vectors containing antigens of HPV16 has shown that IP and subcutaneous routes induce prominent TEM-like CD8 T cell responses and complete protection against tumour challenge, while intranasal (IN) infection results in weak responses and partial protection [36]. Vector-induced tumour-specific CD8 T cell responses were required to overcome a total circulating CD8 T population percentage threshold to provide full protection. Hence, the administration route appears to be another defining factor for protection by CMV vectors.

Mucosal immunization by CMV vectors may provide better protection against specific targets, because robust virus replication is present in lungs and salivary glands after IN infection [180]. In line with this, IN administration of an MCMV vector expressing the M protein from respiratory syncytial virus (RSV) in the *ie2* gene (MCMV-M) generated a robust and protective inflationary CD8 T-cell populations in the lungs with a higher magnitude than systemic (IP) immunization [30]. IN immunization elicited earlier protective responses against RSV challenge and a specific tissue resident memory (TRM) CD8 T cell response in the lungs. This TRM population was 1000-fold more elevated upon IN than IP immunization and significantly higher than RSV infection. TRM CD8 T cells generated by IN vaccination responded rapidly upon RSV challenge with higher IFNγ and macrophage inflammatory protein-1beta (MIP-1β) production in the lung in comparison to IP immunization, leading to 100-fold lower viral loads. TRM ability to take residence in non-lymphoid peripheral tissues permits a fast and definitive secondary response at the site of pathogen entry [181]. Importantly, MI was required for the maintenance of CD8 TRM cells in the lungs and early viral control and it was dependent on the route of administration [32] (Figure 4). TRM have become a promising objective for immunization and a hot topic in vaccine research following the experience with intranasal administration of a live-attenuated Influenza A Virus (LAIV) vaccine. LAIV vaccine was able to generate long-term virus-specific CD4 and CD8 TRM in the lungs of mice, which mediate cross-strain protection independent of circulating memory T cells or neutralizing antibodies [182]. Therefore, it is not surprising that an MCMV vector encoding an influenza antigen induced TRM responses and provided protective responses when applied intranasally [35]. Not unexpectedly, TRM activity in cancer immunotherapy had also been recently highlighted [183]. The role of TRM CD8 T cells in improving anti-tumour efficacy has been demonstrated by a recombinant vaccinia virus (rVACV) carrying OVA in the V16-OVA melanoma growth model, as it was found that the generation of skin TRM cells is sufficient for effective antitumor immunity [184]. Further work is necessary to ascertain the differences and commonalities between TRM and TEM but initial research positions TRM as a functional and protective population that can be induced by CMV vectors. Nevertheless, questions about the limits of murine models in TRM research, as well as the potential immunopathogenic effects associated with TRM induction remain to be addressed [185].

## 5. Conclusions

CMV vectors are very attractive candidates for the priming of immune responses to induce protection against a wide range of infectious diseases and neoplasias. Initial excitement has given place to a rational approach, particularly after unexpected results from the RhCMV model that calls for detailed dissection of the immune mechanism behind their effects [186]. While vector hold the promise of outstanding immune protection, important questions remain open regarding biosafety and vector production. A practical approach that focuses on vector effectiveness and safety in experimental models will need to be complemented with laborious technological advances and early clinical testing. Our understanding of the determinants of MI, its protective significance and its potential immunopathogenicity is well-characterized in animal models [16]. The translational constraints of the most advanced CMV candidates now face challenges that can only be definitely answered in first-in-human trials. Even if unsuccessful, such findings will be of extraordinary help in the development of other closely related candidates. The intricate interaction between vector, antigen and vaccine immune response requires a comprehensive system-immunology approach and each experience with a new CMV vector as vaccine candidate contributes to the solution of this three-parts puzzle. Nevertheless, not every disease should be targeted with a CMV-based strategy. CMV provides extremely strong CD8 T cell immunity, where MI harnessing might be an asset in developing vaccines against intracellular pathogens with low immunogenicity. For other disease targets, CMV vectors may work only in dependence of the pathogenic phase they are directed to. A recent study on RhCMV expressing *Plasmodium knowlesi* antigens showed failure to control blood stage parasites, which is antibody dependent [17]. The pathogenic HCMV potential requires the use of attenuated CMV vectors and ideally replication-deficient ones, separating pathogenicity from immunogenicity. Moderate risks are reasonable trade-off against severe emerging diseases where a rapid induction of effector responses is crucial but this balance need to be carefully assessed. Unexpected adverse effects in clinical trials provoke misconceptions and scepticism despite adequate disease protection, ultimately leading to vaccine hesitancy as the Dengaxia controversy has shown [187]. The study of CMV immunology has been a fascinating but often unpredictable enterprise. Most of our knowledge of the immune response against CMV comes from the mouse model, which requires careful interpretation of the results. Nevertheless, T cell immunity in MCMV and HCMV biology show striking similarities, as results initially described in the mouse model of CMV immunotherapy and immunoevasion were highly predictive of subsequent clinical observations of HCMV biology and disease [4]. Hence, it is reasonable to assume that this animal model recapitulates numerous characteristics of the human immune response relevant to vaccine research. Similarly, results from RhCMV studies provide valuable blueprints for the design of HCMV based vaccine vectors. Therefore, we propose that a better understanding of CMV biology and antigen processing will be decisive in the development of safe and effective CMV vectors for human or veterinary diseases. Once a deeper understanding of CMV, host and pathogen biology is achieved, a path for efficient vaccines based on CMV vectors will be clear.

## Figures and Tables

**Figure 1 vaccines-07-00152-f001:**
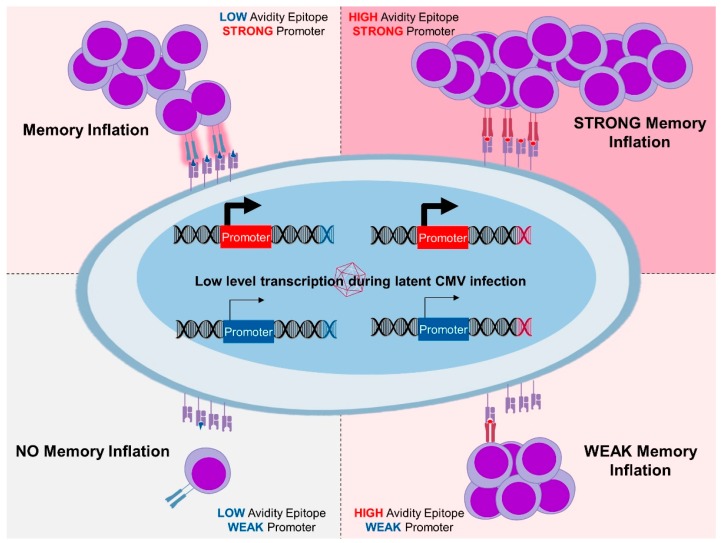
High and low avidity epitopes expressed by strong promoters revealed inflationary response kinetics. Those expressed under weak promoters induced less robust responses, demonstrating that quality of T-cell responses depends on the context of antigenic expression. While the response was always weaker against low avidity antigens, memory inflation was maintained even against them. Weak expression and low avidity resulted in weak responses and no memory inflation.

**Figure 2 vaccines-07-00152-f002:**
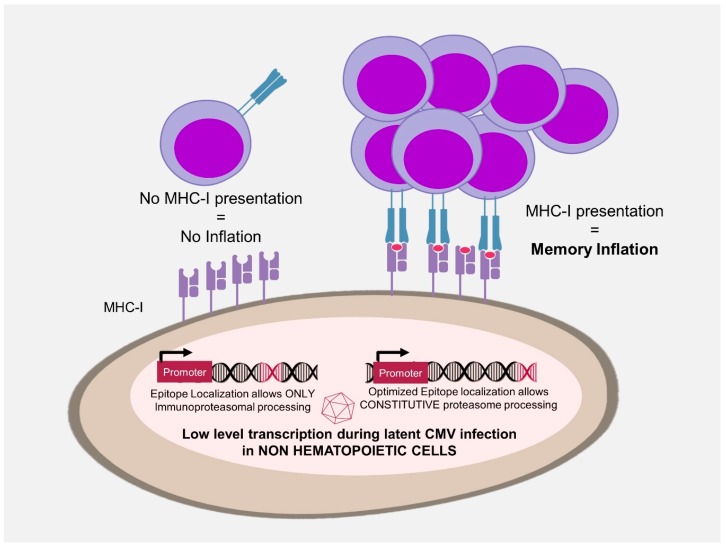
Antigen accessibility and memory inflation. Locating an epitope sequence to the C-terminus of a protein optimizes peptide processing by the constitutive proteasome and allows the development of inflationary CD8 T-cell responses. This acts as a primary filter that acts independently of the promoter expressing the antigen and selects the epitopes that can be presented by major histocompatibility complex I in non-professional antigen presenting cells.

**Figure 3 vaccines-07-00152-f003:**
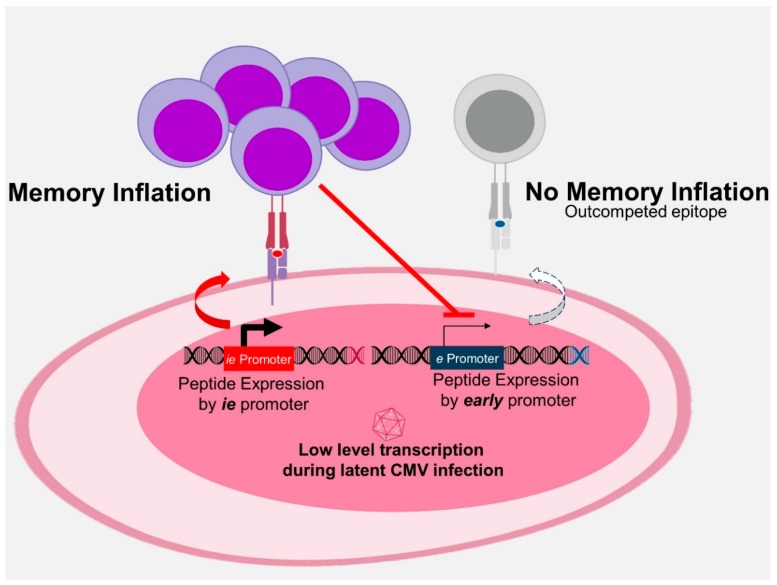
Antigenic competition. Strong and early expression of epitopes engaging T cell receptors with high avidity selects for cognate T-cell clones. These T cells repress subsequent Cytomegalovirus gene expression and thus restrict the MHC presentation of antigenic epitopes that would be expressed later in a reactivating virus. Therefore, such epitopes are progressively outcompeted and the immune response increasingly focuses on the selected few winners of inflation.

**Figure 4 vaccines-07-00152-f004:**
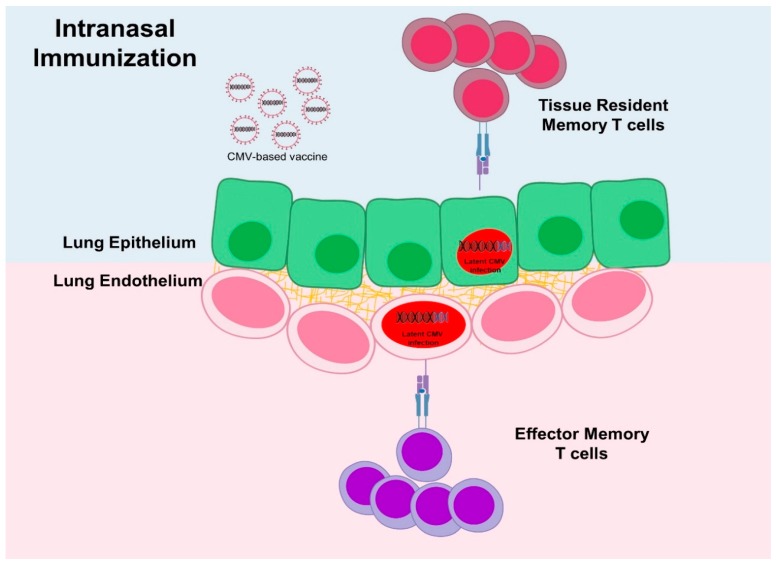
The infection route determines the sites of virus infection and thus the type of T-cell responses. Upon intranasal infection, MCMV reaches cells in lungs on both sides of the basal lamina and induces tissue resident memory T cell (TRM) responses in the surface of the lung epithelium but also reaches stromal cells that may induce effector memory T cell (TEM) responses. Parenteral infection cannot induce TRM, likely due to poor access to the epithelial cells.

**Table 1 vaccines-07-00152-t001:** Protective cytomegalovirus (CMV)-based vaccines against infectious diseases.

Vaccine Candidates	Vector Backbone	Immune Response	Disease Model	Antigen	Design Context	Protection
RhCMV/Pk [17]	RhCMV strain 68-1	Polyfunctional TEM CD8 T cells. Response to non-canonical epitopes with atypical MHC restriction.	*Plasmodium knowlesi* sporozoite challenge	Codon-optimized AMA1, CSP, MSP1c or SSP2 under HCMV gH promoter	Inserted in *Rh211*	3/16 were blood-stage free until day 10 post challenge.
RhCMV/TB set [18]			Erdman strain Mtb intrabronchial challenge	Polyproteins containing Ag85A, Ag85B, Rv3407, Rv2626, Rv1733, ESAT-6, Ag85B, RpfA, RpfC and RpD under MCMV ie promoter, human EF1a promoter or Rh107 promoter., Rpf A and Rpf D polyprotein under the Rh107 promoter	In place of *Rh211* or *Rh107*	14/34 did not have detectable granulomatous disease.
RhCMV/SIV set [19,20,21,22]			Highly pathogenic SIV intra-rectal challenge	Env, gag, rev-tat-nef (fusion construct), pol1 and pol2 under human EF1a promoter or HCMV gH promoter.	Inserted between *rh213* and *Rh214*	12/24 showed rapid control after infection and long-term protection.
ΔRh110 RhCMV/SIV set [23,24]	Attenuated RhCMV strain 68-1					7/13 showed long-lasting absence of SIV viremia
RhCMV/EBOV-GP [25]	RhCMV strain 68-1	IgG responses correlated to protection with no neutralizing antibodies.	EBOV with lethal dose challenge	Codon-optimized EBOV GP	In place of *Rh112*	3/4 survived EBOV challenge with undetectable viremia
MCMV/ZEBOV-NP_CTL_ [26,27]	MCMV Smith strain with *m157* deletion to avoid NK control	Polyfunctional TEM-biased ‘inflationary’ CD8 responses	Mouse-adapted ZEBOV IP challenge	H2b-restricted T cell epitope from ZEBOV NP	Fused to the C- terminus of ie2	All were protected against lethal challenge
RAE-1γMCMVList [28]	Attenuated RAE-1γMCMV	Protective CD8 T cells	*Listeria monocytogenes* *EGD strain (serovar1/2a)*	Listeriolysin O _91–99_ CD8 T-cell epitope	In place of the immunodominant m164_167–175_	All vaccinated mice survived after challenge.
MCMV-85A [29]	Δm1-16-FRT-MCMV: deletion of MHC class I downregulators	CD8 T cell response after AdV vector booster.	Erdman strain Mtb intranasal challenge	Mtb 85A expressed under HCMV ie promoter	In place of backbone deletion	Lung mycobacterial load reduction
MCMV-M [30]	MCMV K181 strain with *m157* deletion [31]	IN route induced inflationary CD8 TEM and TRM in the lungs.	RSV IN challenge	RSV M protein	Inserted in *ie2*	IN route had lower lung viral loads than IP. MCMV-M2 failed to mediate early RSV control.
MCMV-M2 [32]		IN route induced a non-inflationary response.		RSV M2 protein		
MCMV^ie2SL^ MCMV^M45SL^ [33]	MCK2-repaired MCMV Smith Strain [34]	MCMV^ie2SL^ Inflationary CD8 T cell response MCMV^M45SL^ Non-inflationary CD8 T cell response	HSV-1 challenge on susceptible 129/Sv mice	K^b^-restricted peptide from HSV-1 glycoprotein B_498–505_	MCMV^ie2SL^ in-frame fusion to the C-terminus of ie2 MCMV^M45SL^ in the 3′ end of *M45*	No viral detection by day 7 post-challenge in brains and lungs
(HA)-MCMV^IVL^ [35]		IN route induced lung CD8 TRM accumulation	IAV PR8M variant IN challenge	MHC-I restricted peptide IVL_533–541_	Inserted into the C-terminus of the ie2	IN reduced lung viral load and weight loss at a higher magnitude than IP route
MCMV^ie2E6/E7Full^ [14]		Protective CD8 T cells	Heterotopic (subcutaneous) administration of TC-1 cells transformed with HPV16 E6 and E7-oncogenes	Full-length HPV16 E6 and E7	Antigen fused to the C-terminus of ie2	Transient limitation of tumour cell growth
MCMV^IE2E7^ [14]				MHC-I restricted HPV16 E7_49–57_ epitope	Epitope fused to the C-terminus of ie2	No tumour cell growth upon challenge
MCMV-M79-FKBP-E7 [36]	MCMV Smith Strain with FKBP-mediated destabilization of the essential M79 gene					

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
