# Peer review of "Vaccine Vectors Harnessing the Power of Cytomegaloviruses"

_vaccines, 2019, doi:10.3390/vaccines7040152_

Round 1
Reviewer 1 Report
In this review article the authors consider the potential power of CMV as a novel vaccine platform against viral and bacterial pathogens as well as possible use as an anti-cancer vaccine strategy. The review covers in depth various aspects of CMV immune responses/ vaccine platforms in mouse and NHP models with MCMV and RhCMV respectively and this is covered in great detail with a number of interesting points. However, the review is overly long with over 200 references cited. The article would benefit from significant editing to reduce length and focus on main aspects that fits in with the general theme of the article. A summary table of recombinant CMV vaccine strategies against various pathogens/success etc would be beneficial. Unfortunately, the authors fail to evaluate significant hurdles that potentially limit the utility of a CMV vaccine platform strategy (regardless of the plus side) in the context of going from preclinical animal models to clinical trials. The most obvious issue is that the current and highest priority for a CMV vaccine is actually against congenital CMV (cCMV). No published vaccine strategy currently prevents cCMV and consequently it would be unlikely that a CMV vaccine platform could in reality be explored unless this goal was attained. Indeed, there is only one animal model that permits HTP evaluation of cCMV vaccines and success in this model is not mentioned or discussed. Linked to cCMV is the enigma of natural CMV infection, seropositive status and disconnect of protection against cCMV. In essence, it is unclear what aspect of the immune response enables complete protection against cCMV and this continues to be a significant focus of discussion. Additionally, there is an undefined link of CMV to cancer and so it is unclear if a CMV vaccine platform strategy has the potential to increase this risk or indeed if the authors consider CMV & cancer a significant problem/complication. Additionally, CMV also has links to immunosenescence in the elderly -what risks might a CMV vaccine platform in this context or can this be avoided by re-tweaking of the vaccine platform. Furthermore, from a production and distribution standpoint, a CMV vaccine platform strategy has additional obstacles not seen with other live attenuated viral vaccines and this aspect should be addressed. Potentially, some of these points could be raised in the final conclusions section of the review and/or the introduction section. The authors should also raise the potential additional complications of studies based on MCMV where there are fundamental differences between HCMV and MCMV and also their specific hosts. Humans encode IL8 but mouse does not and additionally HCMV encodes IL8 homologs but against MCMV does not. HCMV encodes a potent IL10 homolog and MCMV does not -what impact does this have on the utility of the mouse model for informative studies that are directly translational. The authors severely overstep the level of success of CMV vaccine strategies against TB without any counter balance (eg. A85 antigen might have worked in mice but has failed in human clinical trials as an effective protective antigen but this fact is not noted). Similarly, the authors suggest that a CMV vaccine strategy against Ebola will be more effective than a VSV recombinant approach. Considering that a VSV based Ebola vaccine has 97% efficacy after 1 inoculation and is protective within 6 days of vaccination a statement of CMV vaccine strategy being better would seem a major leap without fact. Finally, faced with an increasing trend against vaccines by certain populations in US and world-wide how does a CMV vaccine platform rise above this problem. The authors mention Dengvaxia (a recombinant viral vaccine against DV) and clinical trials with this vaccine are currently steeped in problems and an obvious issue to discuss- perhaps in conclusions section.
Specific points
L30-31. Both live attenuated and inactivated vaccines have been used against Polio. Indeed it could be argue that the ease of vaccination and hence rapid herd protection with the Sabin vaccine could be attributed to the near world wide eradication of Polio.
L32 What specific severe infections? Table?
L33 Chronicity? It is unclear if this is in reference to presence in a specific population (endemic) or is a chronic infection of a specific host? A specific example would be helpful to covey the point.
L41 compliance- please elaborate.
L48 “particularly interesting”? All viruses are interesting- clarify.
L78 MVA can be grown on mammalian cells- BHK-21 and also a specific Rat cell lines in addition to chicken fibroblasts.
L84 Denvaxia- recombinant DV vaccine strategy and associated clinical trial deaths and associated mistrust with Measles vaccine should be discussed in conclusions section for a contrast to CMV vaccine platform against any pathogen.
L87 Ad5- other strains of Ad which are less common are being explored as well as the use of Ad DISC vaccines.
L102/103 What is the pathway to CMV reactivation and does this present an issue for a CMV vaccine
L130-L140 MCMV does not encode a PC- how does this impact on the translation nature of MCMV studies? A non-replication competent DISC GPCMV vaccine strategy failed to protect against cCMV- how does this impact on the potential for a DISC vaccine strategy being used as a CMV vaccine platform?
L149-152 pp150 (UL32) is essential in HCMV and so a RhCMV atten vaccine strategy does not translate for HCMV vaccine approach – please comment.
L202-204 85A is not a protective antigen against TB in humans.
L208-215 m157 is not encoded by HCMV and m157/Ly49 NK cell is a associated with c57 black background mice/MCMV infection. How does this compare to balb/c background for immune response?
L217-268 It should be made clear that these genes are encoded in MCMV but not HCMV- what would be equivalent in HCMV?
L365 Chronic? Persistent?
L400 MMCMV typo
L441 ie2 is not homolog of IE2 HCMV, which is ie3 in MCMV- please clarify. Specific differences exist between major immediate early locus of HCMV and MCMV raise against specific translation issues (eg. IE1 essential in HCMV but not in MCMV).
L459 M45 is slightly more than just an early gene and M45 functional protein associated with anti-necroptosis is not present in HCMV.
L472 HCMV and link to metastatic cancer should potentially be discussed here if evaluating vaccines against cancer.
L744-773 This section should be revised to encompass additional aspects not covered in the review that also might be roadblocks to successful use of a CMV vaccine platform see above section.
Author Response
Responses to Reviewer #1
Reviewer #1 has praised the depth of the coverage of the area by our review, but has raised several critical points. We thank the insightful commentaries and recommendations and we provide a point by point reply on how we addressed them:
”…the review is overly long with over 200 references cited. The article would benefit from significant editing to reduce length and focus on main aspects that fits in with the general theme of the article. A summary table of recombinant CMV vaccine strategies against various pathogens/success etc would be beneficial”
We have shortened within reason the article (approximately, 4 pages) and focused our attention on the infectious application of the CMV vectors. We have reduced the information on basic immune mechanisms that are not critically relevant to the understanding of the use of CMV as vaccine vectors. We included a table containing relevant information from CMV candidates against infectious diseases, describing their strategies and construction.
“The authors fail to evaluate significant hurdles that potentially limit the utility of a CMV vaccine platform strategy (regardless of the plus side) in the context of going from preclinical animal models to clinical trials. The most obvious issue is that the current and highest priority for a CMV vaccine is actually against congenital CMV (cCMV) No published vaccine strategy currently prevents cCMV and consequently it would be unlikely that a CMV vaccine platform could in reality be explored unless this goal was attained (...)”
We politely but firmly disagree with the reviewer that we disregarded the vector virulence aspect. cCMV pathogenicity requires productive viral replication and we described in detail several strategies used by various labs to develop attenuated or replication-deficient CMV vectors, which retain immunogenicity but show less virulence. Nevertheless, we have introduced a specific remark to point out that these strategies are likely to address concerns about cCMV disease as well. (see line 111 and the following two paragraphs in the section on initial considerations)
“…there is an undefined link of CMV to cancer and so it is unclear if a CMV vaccine platform strategy has the potential to increase this risk or indeed if the authors consider CMV & cancer a significant problem/complication.”
We agree with the reviewers and we included this part on the section “Initial considerations” (lines 149-168).
“CMV also has links to immunosenescence in the elderly -what risks might a CMV vaccine platform in this context or can this be avoided by re-tweaking of the vaccine platform.
The role of CMV in immunosenescence or inflammatory disease is unclear, but we have included a paragraph in the chapter on initial considerations (lines 169-180).
Furthermore, from a production and distribution standpoint, a CMV vaccine platform strategy has additional obstacles not seen with other live attenuated viral vaccines and this aspect should be addressed.”
We consider that including production and distribution of CMV vectors is outside the scope of this already lengthy review. Nevertheless, we have added a brief note on this aspect in line 96-97.
“The authors should also raise the potential additional complications of studies based on MCMV where there are fundamental differences between HCMV and MCMV and also their specific hosts”
The limitations of animal models towards clinical applications, is referred to in the concluding paragraph (lines 667-675).
”The authors severely overstep the level of success of CMV vaccine strategies against TB without any counter balance. Similarly, the authors suggest that a CMV vaccine strategy against Ebola will be more effective than a VSV recombinant approach.”
We have now restricted the description of MCMV vectors encoding the TB antigen, as a curious example of a vector that provided antigen-independent, NK-cell based immunity against TB (lines 242-248). The RhCMV-TB vectors are described only in comparison to RhCMV vectors providing protection against SIV and to illustrate the epitope classes that provide protection against various pathogens (lines 508-510). We caution that additional studies are required to evaluate the translational potential for the animal studies (lines 650-652)
With regards to the Ebola vaccine using the VSV approach, we highlight it in the introduction as a breakthrough example of a viral vector that reached clinical application (lines 71-73). VSV-based vaccine VSVΔG/ZEBOVGP is mentioned as the standard that was used to compare MCMV vectors in the ZEBOV mouse model, and MCMV/ZEBOV-NPCTL is described as comparable, but not better (lines 381-388). The introduction of CMV as potential vaccine vector was redacted to reflect its potential, but avoid wording that may be interpreted as overpromising (lines 74-77).
“Finally, faced with an increasing trend against vaccines by certain populations in US and world-wide how does a CMV vaccine platform rise above this problem. The authors mention Dengvaxia (a recombinant viral vaccine against DV) and clinical trials with this vaccine are currently steeped in problems and an obvious issue to discuss- perhaps in conclusions section.”
We very much agree with the reviewer regarding the need of an open discussion around the Dengvaxia controversy and vaccine hesitancy, and we added a commentary in the conclusions (lines 665-667). Nevertheless, for reasons of length and scope, we did not discuss this in depth.
Specific points
L30-31. Both live attenuated and inactivated vaccines have been used against Polio. Indeed it could be argue that the ease of vaccination and hence rapid herd protection with the Sabin vaccine could be attributed to the near world wide eradication of Polio.
This section has been deleted in line with recommendations to shorten the manuscript by both reviewers.
L32 What specific severe infections? Table?
We have listed the infections not currently targeted by efficient vaccines according to an official source (line 28-31). A table appears outside of the scope of this review, which focuses on CMV vectors.
L33 Chronicity? It is unclear if this is in reference to presence in a specific population (endemic) or is a chronic infection of a specific host? A specific example would be helpful to convey the point.
We clarify this and added Mtb as an example (lines 31-34).
L41 compliance- please elaborate.
This sentence appeared non-essential, and we eliminated it to sharpen the focus of the review in line with the general recommendation by both reviewers.
L48 “particularly interesting”? All viruses are interesting- clarify.
We have altered the wording, eliminated the overly broad moniker “particularly interesting” and described the properties that make viral vaccine vectors interesting tools for the induction of protective immune response (lines 36-38).
L78 MVA can be grown on mammalian cells- BHK-21 and also a specific Rat cell lines in addition to chicken fibroblasts.
This sentence was also eliminated, because this level of detail was out of the scope and focus of our review.
L84 Denvaxia- recombinant DV vaccine strategy and associated clinical trial deaths and associated mistrust with Measles vaccine should be discussed in conclusions section for a contrast to CMV vaccine platform against any pathogen.
We described briefly the Dengaxia controversy and its impact on vaccine mistrust (lines 665-667), but we think that expanding on this aspect would go beyond the scope of our review.
L87 Ad5- other strains of Ad which are less common are being explored as well as the use of Ad DISC vaccines.
We have now included AdV strains with lower prevalence and Chimpanzee AdV as strategies to overcome preexisting immunity (lines 51-57).
L102/103 What is the pathway to CMV reactivation and does this present an issue for a CMV vaccine
We added a seminal paper on reversion to virulence from live attenuated CMV vaccine (lines 133-135) and further described how attenuation is necessary to diminish this risk (entire third paragraph of the section 2 – Initial considerations).
L130-L140 MCMV does not encode a PC- how does this impact on the translation nature of MCMV studies?
The presence or absence of the pentameric complex in HCMV infection does not appear to alter the quality of CD8 T-cell responses, which remain MHC-I restricted regardless of the context (lines 513-516). Therefore, the MHC-I restricted CD8 responses that were studied in the murine model appear to be a valid reflection of immune responses to HCMV infection and the absence of the pentameric complex does not affect this outcome.
L130-L140 A non-replication competent DISC GPCMV vaccine strategy failed to protect against cCMV- how does this impact on the potential for a DISC vaccine strategy being used as a CMV vaccine platform?
A non-replicating DISC GPCMV significantly diminished the percentage of pups that were GPCMV positive upon challenge with a replication competent GPCMV (Choi et al. J. Virol. 2016). However, this study did not test whether replication incompetent GPCMV used as vaccine caused any brain pathology in pups. Therefore, the study in question does not provide insights relevant for the use of CMV based vectors. We added a sentence in the introductory remarks to highlight that the focus of this review is not on vaccines against CMV itself (lines 77-79)
L149-152 pp150 (UL32) is essential in HCMV and so a RhCMV atten vaccine strategy does not translate for HCMV vaccine approach – please comment.
We added a sentence (lines 146-148) to highlight that replication incompetent RhCMVs or MCMVs are used as a general indication that similar decoupling of pathogenicity and immunogenicity might also be observed in HCMV vectors.
L202-204 85A is not a protective antigen against TB in humans.
We agree that the protectivity of TB 85A has not been demonstrated in humans (Lancet 2013), although it is not immediately clear if the lack of protection was due to the antigen or to the vector. However, we referred to the MCMV vector expressing the 85A antigen to illustrate an example of an NK-cell dependent protection, where antigenic properties of 85A may have been secondary for the protection. We have emphasized these points to avoid confusion (lines 242-248 of the revised manuscript).
L208-215 m157 is not encoded by HCMV and m157/Ly49 NK cell is a associated with c57 black background mice/MCMV infection. How does this compare to balb/c background for immune response?
This section was considered non-essential and removed. Inflationary T-cell responses were initially described in BALB/c mice (Holtappels et al. J. Virol. 2000, 2002) and subsequently shown to occur in C57BL/6 mice as well (Munks et al. J. Virol 2005). Therefore, the strain differences in NK cell responses do not seem to affect the ability of CMV vectors to provide T-cell mediated protection. We consider this a minor point and thus did not elaborate on it.
L217-268 It should be made clear that these genes are encoded in MCMV but not HCMV- what would be equivalent in HCMV?
We added the description of the HCMV vector expressing the human ligand ULBP2 for the human NKG2D receptor on NK cells and data showing that this vector behaved in a similar fashion in humanized mice (238-241).
L365 Chronic? Persistent?
Viral persistence was mentioned in lines 360, 364 and 367. We added a description at first occurrence to specify that we refer to “lifelong intermittent antigen expression in latency” (line 310)
L400 MMCMV typo
The typo has been corrected. Thank you for pointing it to us.
L441 ie2 is not homolog of IE2 HCMV, which is ie3 in MCMV- please clarify. Specific differences exist between major immediate early locus of HCMV and MCMV raise against specific translation issues (eg. IE1 essential in HCMV but not in MCMV).
We added a sentence and a citation to highlight that epitopes from HCMV ie1 gene are also immunodominant (lines 376-377). Therefore, the CD8 response to ie epitopes described in MCMV infection appears to be a reflection of the human situation. Hence, the various differences between sequence, structure and function of ie genes in HCMV and MCMV are irrelevant for the study of T-cell responses to ie antigens in the MCMV model.
L472 HCMV and link to metastatic cancer should potentially be discussed here if evaluating vaccines against cancer.
We have added a paragraph on the putative role of CMV in cancer to the “Initial Considerations” chapter (lines 149-168) to discuss this aspect in a more generic manner.
L744-773 This section should be revised to encompass additional aspects not covered in the review that also might be roadblocks to successful use of a CMV vaccine platform see above section.
We have altered this section (lines 642-679) and in line with reviewer’s recommendation.
Reviewer 2 Report
The review by Ynga-Durand et al. carefully summarizes the current understanding of immunological responses to CMV infection with emphasis on using CMV as a vaccine vector. The article is well written, comprehensive, and timely. To the best of this reviewer's ability, the article accurately reflects the primary literature. There are several suggestions for the authors to consider to improve the impact, focused on reducing the density of information.
Section 1 goes beyond the scope of the article's intentions and should be more focused. Although the section is well written, focusing the information will help the reading integrate the incredible about of information presented in the remainder of the article. The extraordinary differences in immunological responses between CMV positive and negative individuals should receive more attention with a detailed discussion of studies from the laboratory of Mark Davis. Although the article is comprehensive, the information is difficult to integrate due to its complexity and density. The article would be significantly improved by including a table summarizing the virus background (e.g. mutation, encoded antigen), type/site of inoculation and the resulting immunological response divided into categories. The current figures are marginally helpful. Additional proofreading is needed.
Overall, this is clearly written review and likely to be well received.
Author Response
Responses to Reviewer #2
We appreciate Reviewer #2 commentaries and suggestions. We made the following modifications:
"Section 1 goes beyond the scope of the article's intentions and should be more focused."
We reduced its extension and delimitate the scope of our review. In particular we shortened the initial section and the section on innate immunity.
"The extraordinary differences in immunological responses between CMV positive and negative individuals should receive more attention with a detailed discussion of studies from the laboratory of Mark Davis"
We rectified this omission and included this groundbreaking study in the section “Initial considerations” as an example of favorable CMV effects in immunocompetent individuals (Lines 175-177).
"The article would be significantly improved by including a table summarizing the virus background (e.g. mutation, encoded antigen), type/site of inoculation and the resulting immunological response divided into categories."
We added a table with the CMV vectors that showed protection in infectious models of disease, describing their design and highlighting their immunology.
"Additional proofreading is needed."
The entire manuscript was corrected and a simplified, clearer language was used in all sections. These changes were too numerous to indicate in all of the sections.

Reviewer 3 Report
The present review paper “Vaccine vectors harnessing the power of Cytomegaloviruses”, submitted by the Mario Alberto, Ynga-Durand, Iryna Dekhtiarenko, and Luka Cicin-Sain has described an accurate and detailed report on the CMV as a vaccine vector. The authors provided a concise introduction on various vaccine vectors including vaccinia virus, Lentivirus, Adenovirus and others. Moreover, the authors also described the replication-defective virus as a suitable vaccine vector. Very interestingly, the authors have provided sufficient examples of CMV as the vaccine vectors for TB, HIV, malaria and many other diseases in a table format. The authors described that CMV could be an excellent vaccine vector. To this end, authors have provided an elaborative description on CMV specific innate immunity, adaptive immunity including B-cells and T-cells based immunities. To more details, authors have included unconventional MHC restriction and CD4 T cells for the CMV encoded epitopes. Several factors including doses with administration route effect the efficacy of a vaccine. In the end, the authors have well described the route of CMV administration and their respective outcomes. This paper is well written without any flaws and has sufficient information in regard to the CMV vaccine vectors and other details and should be accepted in the journal.
Minor comments
A recent study has reported CMV as a vaccine vector. This paper should be cited in this review paper.
Liu J, Jaijyan DK, Tang Q, Zhu H. 2019. Promising Cytomegalovirus-Based Vaccine Vector Induces Robust CD8(+) T-Cell Response. Int J Mol Sci 20.
Author Response
We thank the reviewer for his kind words.
The review in question has indeed eluded our attention so far, but we added it to the MS.
Reviewer 4 Report
This is a well-written and comprehensive review. The authors have carefully described both the pros and cons of using cytomegalovirus as a vaccine vector. I believe that it will be of great interest to the readers of Vaccines.
Author Response
We thank the reviewer for his kind words and positive review.